# Promoting the Participation of “Hard-to-Reach” Migrant Populations in Qualitative Public Health Research during the COVID-19 Pandemic in Chile

**DOI:** 10.3390/ijerph20031956

**Published:** 2023-01-20

**Authors:** Alice Blukacz, Báltica Cabieses, Alexandra Obach, Alejandra Carreño Calderón, María Inés Álvarez, Paula Madrid, Isabel Rada

**Affiliations:** 1Programa de Estudios Sociales en Salud, Instituto de Ciencias e Innovación en Medicina, Facultad de Medicina Clínica Alemana, Universidad del Desarrollo, Avenida Plaza 680, Las Condes, Región Metropolitana, Santiago 7610658, Chile; 2Department of Health Sciences, University of York, York YO10 5GH, UK

**Keywords:** public health research, qualitative research, hard-to-reach populations, international migrants, COVID-19 pandemic, Chile

## Abstract

The COVID-19 pandemic has further deepened socioeconomic and health inequities worldwide, especially among populations experiencing social vulnerability, such as international migrants. Sustained lockdowns and social distancing have raised challenges to conducting public health research with hard-to-reach populations. This study aims at exploring strategies to recruit “hard-to-reach” international migrants for qualitative public health research during the pandemic in Chile, based on the authors’ experience. A retrospective qualitative evaluation process was carried out on the recruitment processes of three qualitative research projects focused on international migrants in Chile. All projects were implemented during the COVID-19 pandemic, demanding complementary and flexible strategies: (i) social media; (ii) snowball sampling; (iii) referrals from social workers and pro-migrant and migrant-led organizations; (iv) vaccination centers and healthcare centers; and (v) community-based recruitment. The strategies are qualitatively evaluated around seven emerging qualitative categories: (i) feasibility during lockdown periods; (ii) speed of recruitment; (iii) geographical coverage; (iv) sample diversity; (v) proportion of successful interviews; (vi) ethical considerations; and (vii) cost. Engaging hard-to-reach international migrants in public health research during the pandemic required constantly adapting recruitment strategies. Furthermore, relying on strategies that were not only Internet-based promoted the participation of populations with limited access to the Internet and low-digital literacy.

## 1. Introduction

The ongoing COVID-19 pandemic has further deepened socioeconomic and health inequities worldwide, especially in highly unequal countries such as Chile [1]. As a population group that has historically faced social inequalities, and considering that migration is recognized as a social determinant of health, international migrants are a public health and medical research priority during the pandemic [2]. According to the International Organization for Migration, an international migrant is someone who “moves away from his or her place of usual residence across an international border, temporarily or permanently, and for a variety of reasons” [3].

Chile had traditionally received labor migrants from neighboring Peru [4]; however, migratory flows are becoming increasingly diversified. The top country of origin of the 1,462,103 international migrants residing in Chile at the end of 2020 was Venezuela, with 448,865 people [5]. Although Peruvian migrants remain a highly represented group, they are now closely followed by migrants from Colombia and Haiti. Latin American migrants in Chile experience higher rates of multi-dimensional poverty than their local counterparts [6], and an estimated 16.3% remained unaffiliated with the healthcare system in 2017 [7]. Additionally, recent inflows of undocumented migrants traveling by foot and entering Chile through unauthorized crossing points are unaccounted for, which suggests that this rate may be higher [8]. These groups also experience extreme socioeconomic, psychosocial, and legal vulnerability linked to their precarious migratory status [8,9].

Measures to hinder community transmission of SARS-CoV-2 have commonly included mobility restrictions such as lockdowns. In Chile, as of 15 May 2020, more than 40% of the 19 million residents lived under strict lockdown [10], and although mobility limitations were eased or tightened according to incidence and mortality rates, in March 2021, 80% of the Chilean population were again living under lockdown [11]. Some districts, among the most socioeconomically vulnerable in the country, remained under total or partial lockdown for months at a time [12], and the curfew established in March 2020 was only lifted on 1 October 2021 [13]. At the pandemic’s peak in Chile in June 2020, nearly 60,000 cases were active country-wide before a second peak in early June 2021 with 50,000 active cases. However, numbers fell steeply in mid-June 2021, mainly due to mass vaccination [14]. In early September 2021, the country had fewer than 3000 active cases, the lowest number since the end of March 2020 [15], which led to reducing mobility restrictions. All measures related to COVID-19 were lifted in October 2022.

Face-to-face community engagement, recruitment, and data collection are critical in public health research [16,17,18]. However, sustained lockdowns and recommendations to limit interactions with the public outside individual households, or to carry out essential activities, have raised a significant challenge to onsite recruitment of participants and data collection [19,20,21,22]. In addition, preventing infections and preserving the health of participants, researchers, and their families is a priority. These limitations are exacerbated for hard-to-reach populations [23], defined as populations challenging to contact, engage with, and involve in health research [24], with no straightforward way to contact and recruit them [25]. Barriers to engaging hard-to-reach populations in health research include limited access to or exclusion from healthcare systems [17], geographical location, socioeconomic situation [24,26], high mobility [27], distrust in institutions [26] as well as language and cultural gaps. International migrants have been considered “hard-to-reach” populations by some international experts because they usually meet several or all these criteria and may be reluctant to engage with researchers due to concerns surrounding confidentiality, fear of deportation, or other negative impacts on their status [24,28,29,30,31]. Additionally, when a population’s exact size is unknown, it can be considered “hidden” [25], which may be the case of undocumented migrants, for whom mainly estimated numbers are produced.

In this context, and considering the limitations brought by the pandemic, recruiting participants for qualitative research without in-person interaction may raise the following challenges: recruiting within the boundaries of the researchers’ networks, heavily relying on snowball sampling, difficulties in building trust with potential participants and convince them to participate, administering informed consent forms and engaging with populations who may have low digital literacy and poor Internet access. Regarding remote recruitment strategies used for hard-to-reach populations, social-media and Internet-driven recruitment have become prevalent for being easy to manage, cost-effective, and being able to reach more participants across a greater geographical range [32,33]. These strategies may be complementary to other strategies, such as offline media (leaflets, posters, etc.), face-to-face recruitment strategies in healthcare facilities or community settings [34], and referrals from healthcare providers [35,36].

This study aims at exploring strategies to recruit “hard-to-reach” international migrants for qualitative public health research during the pandemic in Chile, based on the authors’ experience. Including international migrants in Chile and elsewhere in public health research is crucial during this global socio-sanitary crisis, as health inequities must be researched and addressed for both the local and the migrant populations to promote population health. Furthermore, as the pandemic has brought unprecedented challenges to conducting face-to-face research, documenting experiences involving hard-to-reach populations is of interest to researchers in Latin America and other countries with people experiencing socioeconomic vulnerability.

## 2. Methods

This article is based on the authors’ experience conducting research with migrant populations during the pandemic, either as principal investigator, co-investigator, or research project coordinator. All three projects focused on international migrants in Chile and were carried out during the COVID-19 pandemic. The research team engaged in a retrospective reflection process on recruiting hard-to-reach migrant participants as the pandemic pushed us to re-think how to conduct research while complying with physical distancing recommendations. The research question was: What strategies can be implemented to successfully recruit hard-to-reach international migrants to participate in public health research during the COVID-19 pandemic in Chile?

The process was carried out in five steps. First, field notes, meeting reports, and email exchanges between the members of the respective research teams, including principal investigators, research coordinators, recruiters, and interviewers, were reviewed. Second, the authors had a meeting during which each recruitment strategy was discussed in detail. The topics of feasibility, speed of recruitment, geographical coverage, sample diversity, success rate, ethical considerations, and cost emerged from the discussion. Third, AB systematized the qualitative evaluation results around the emerging topics. Fourth, the document was shared with the other authors, who reviewed and modified the draft when needed for accuracy, and any discrepancy was discussed and resolved by the whole team. Fifth, the final draft was reviewed and validated by all the authors.

The characteristics of the three research projects are summarized in Table 1 below.

### 2.1. Research Project 1

The main objective of the first study, titled “International migrant communities’ vulnerability and resources during the COVID-19 pandemic in Chile: Building cross-cultural targeted coping strategies”, was to explore the psychosocial and socioeconomic vulnerability of international migrants in Chile as well as the resources and social capital used by migrants at the individual and community level to cope with the impact of the COVID-19 pandemic [9]. The study was descriptive and exploratory and was a case study under the qualitative paradigm.

The study was carried out entirely remotely between November and December 2020. The sampling was purposive as it sought to include participants from the five top countries of origin living in Chile: Venezuela, Peru, Colombia, Haiti, and Bolivia living in Santiago, Antofagasta, and Arica. Additionally, it sought to include beneficiaries of both the public and private health systems and to find an equal representation of male and female participants. The only exclusion criteria, apart from not being an international immigrant living in Chile, were being under 18 and not speaking Spanish.

Recruitment of the 30-participant sample was carried out with the support of the pro-migrant partner organization Servicio Jesuita a Migrantes, which contacted potential participants among its beneficiaries and shared the phone number of those who gave their consent and showed interest in participating with the research team. Additionally, the research team used their personal networks to recruit participants and diversify the sample. Finally, the sample size was completed through snowball sampling. Individual semi-structured interviews lasted 30 to 45 min and were conducted via telephone or a video call via Zoom or WhatsApp. The interviews were conducted according to a semi-structured interview guide designed by the research team in collaboration with key experts.

### 2.2. Research Project 2

The “International migrants in quarantine facilities in Chile during the COVID-19 pandemic: towards an ethical response to deeply rooted challenges” study aimed at analyzing the experience of international migrants, health professionals and authorities, and facility managers in quarantine facilities in Chile during the COVID-19 pandemic, to enhance an ethical care approach in quarantine facilities as a public health response to pandemics in contexts of human mobility, cross-cultural health, and social vulnerability [8]. The study was descriptive and exploratory and was a case study under the qualitative paradigm.

Participants were recruited, and data were collected during January 2021, entirely remotely. The sampling was purposive as it specifically targeted Spanish-speaking international migrants over 18 years old who currently were, or had previously been, in a quarantine facility in Chile during the pandemic. The regions considered were the Metropolitan Region of Santiago and the Northern regions of Arica y Parinacota, Tarapacá, or Antofagasta. Reasons for being housed in a quarantine facility ranged from testing positive for COVID-19, having been in close contact with someone who tested positive, or having entered the country through an unauthorized crossing point as the border was closed at the time of data collection. In that sense, some participants had an irregular migratory status when interviewed, and great caution was taken to ensure the data were anonymized and confidential, as described in the ethics subsection below. This also meant that accessing this specific population was especially challenging. The participants were recruited through key contacts in pro-migrant and immigrant-led organizations in the Metropolitan Region and the Arica y Parinacota, Tarapacá, and Antofagasta regions. Additionally, an invitation to participate in an interview was shared by the pro-migrant network Infomigra on Instagram. Finally, snowball sampling was implemented to complete samples and reach saturation.

Individual semi-structured interviews were carried out, lasting 30 to 45 min, and were conducted through telephone or a video call via Zoom or WhatsApp. The interviews were conducted according to an interview guide designed by the research team in collaboration with key experts.

### 2.3. Research Project 3

The “Developing capacities for cross-cultural encounters between health teams and international migrants in Chile” research project aims at answering the two following research questions: (i) How is the cross-cultural health model understood and used by patients, health professionals, and health authorities in Chile, focusing on the international migrant population? (ii) What is the immediate effect of a pilot training of cross-cultural skills for health professionals, with a focus on the international migrant population, based on an updated proposal of the current conceptual model? The project seeks to establish a new approach to cross-cultural health specific to the international migrant population in Chile, according to the perspective of health professionals, patients, and health authorities. The study was descriptive and exploratory and was a case study under the qualitative paradigm.

The project was carried out in the Metropolitan Region of Santiago (borough of La Granja), Antofagasta region (borough of Antofagasta), Valparaíso region (borough of Valparaíso), and Araucanía region (borough of Temuco). Each borough considers two primary healthcare centers and a hospital, as well as the resident immigrant population and health authorities at the national level (Ministry of Health and health services).

Due to health professionals’ increased workload and lockdowns in place at national and regional levels because of the pandemic, the start of the project was delayed from March 2020 to March 2021, allowing for onsite recruitment and data collection strategies to be implemented when contagion rates started decreasing. As a result, participant recruitment and data collection began in August 2021 in healthcare centers with highly trained recruiters and interviewers and finished in August 2022. 

### 2.4. Ethical Aspects of the Projects

All three studies included in the article followed the relevant guidelines and regulations for research involving human beings, including the Declaration of Helsinki. All participants provided written informed consent before enrollment in the study by filling out an informed consent form available online through Google Forms or Alchemer before taking part in the interview or survey. Additionally, they could withdraw from the study at any point and refuse to respond to any of the questions.

The first study was approved by the Comité Ético Científico de la Facultad de Medicina Clínica Alemana Universidad del Desarrollo (number 2020-117), the second by the Comité Ético Científico de la Facultad de Medicina Clínica Alemana Universidad del Desarrollo (number 2020-145), as well as the Research Ethics Review Committee of the World Health Organization (number CERC.0064), and the third project was approved by the Comité Ético Científico de la Facultad de Medicina Clínica Alemana Universidad del Desarrollo (number 2019-94).

The privacy and anonymity of all participants were safeguarded. As the projects involved recording interviews, each audio recording was given a code name before moving forward to the transcription and analysis, making it impossible to associate the recording with the participant beyond basic information (sex, country of origin, region of residence). Only the interviewer, project lead researchers, and project coordinator had access to the names of the participants, and no name or image will appear on publicly available documents. When quotes were included in subsequent outputs, only the sex, country of origin, and region of residence of the participant appeared. The material produced was 100% digital and shared exclusively among the research team through an online folder locked with a password.

No monetary compensation was offered to the participants; however, they were offered a PDF resource on access to the public healthcare system and the support networks available to international migrants in the country

## 3. Results

All three projects were implemented during the COVID-19 pandemic in Chile, with infection rates and consequent social distancing and mobility restriction measures waxing and waning, demanding complementary and flexible strategies involving remote recruitment or a combination of remote and face-to-face recruitment. Therefore, two remote and three onsite recruitment strategies are discussed:-Remote strategy: social media;-Remote strategy: snowball sampling;-Remote strategy: referrals from social workers and pro-migrant and migrant-led organizations;-Onsite strategy: vaccination centers and healthcare centers;-Onsite strategy: community-based recruitment;

Table 2 below shows which strategies were implemented to recruit participants for each project.

The strategies were discussed following seven emerging categories. The first category, “feasibility during lockdown periods”, refers to whether it was possible to implement the strategy during periods of mobility restriction due to the pandemic. The second, “speed of recruitment”, alludes to whether the strategy led to interviews in a short time frame. The third one, “geographical reach”, is concerned with whether the strategy reached participants in regions other than the Metropolitan Region of Santiago, where the team is located. The fourth category, “sample diversity”, refers to whether the strategy yielded a diverse sample of participants. The fifth category, “proportion of successful interviews”, is concerned with whether the strategy was perceived as leading to a high number of interviews. The sixth one, “ethical considerations”, encompasses the precautions additional to ethical standards established by the research protocol to be taken when implementing the strategy. Finally, the seventh category, “cost”, concerns whether the recruitment strategy requires significant expenditure.

Table 3 summarizes the results of the process and rates each strategy either low, medium, or high, according to the research team’s perception. The rest of the results section presents a narrative for each strategy justifying the qualitative evaluation shown in the table.

### 3.1. Remote Strategy: Social Media

The first two projects used social media platforms Facebook and Instagram to recruit participants, as they were conducted during periods of mobility restriction and lockdown. No paid advertisement was used because of budget restrictions, and this strategy relied on pro-migrant organizations with a high reach on social media, making it a low-cost strategy.

The first project was advertised on the Facebook and Instagram pages of the Servicio Jesuita a Migrantes between November and December 2020. The invitation was posted every week on their Facebook feed and every two weeks on their Instagram feed and stories throughout the recruitment and data collection period. The second project advertised an invitation to participate in the interview via Infomigra’s Instagram stories every two days during January 2021. Both invitations to participate provided a telephone number for potential participants to message or call the research coordinator to express interest in participating and ask for further information. 

Social media was instrumental in ensuring a broad circulation of the invitation in the context of projects conducted simultaneously in Santiago and Northern regions of Chile by the same research team, guaranteeing geographical reach. However, it did not yield a high number of effective participants. Furthermore, advertising the invitation to participate in studies that require an interview to be carried out rather than filling out a self-administered online survey meant that potential participants had to proactively contact the project coordinator to receive further information and decide whether to participate. Among those who did, some stopped responding upon receiving the information or declared not being interested in participating. In this context, this strategy required a longer time frame to reach sample sizes and saturation, with a low success rate. In addition, international migrants in Chile may be less active social media users due to limited Internet connection or low digital literacy, limiting their online activities to looking for specific information or interacting with friends and family. This reduces the number of potential participants who are available or willing to interact and leaves out migrants experiencing higher levels of social vulnerability due to limited access to the Internet, potentially hindering sample diversity. Finally, no specific ethical constraint was identified besides compliance with the established regulations for conducting research with human beings.

### 3.2. Remote Strategy: Snowball Sampling

All three projects also relied on snowball sampling by asking participants to refer friends, family, or acquaintances who might be interested in participating and meet the inclusion criteria. This strategy was highly feasible during lockdowns as it did not require physical contact with potential participants and allowed the team to reach participants in other regions of Chile than the one where they were located.

The main challenge arising from this strategy, especially in the third project, which requires participants to live in a specific borough where not many international migrants live, and most are newly arrived, is that participants did not know anyone meeting the inclusion criteria, had limited social and support networks, or had limited interactions due to sanitary restrictions. Snowball sampling may also lead to a high number of participants with similar characteristics, but asking participants to refer three potential participants from their social networks randomly could promote sample diversity.

Participants may also refer someone without informing them, despite the interviewer emphasizing the importance of doing so, leading to the person not answering the phone to an unknown number or refusing to participate. However, when the initial participant did so, contact with the referee could be made quickly and usually led to an effective interview in a short time frame.

Regarding ethical considerations, it is crucial to ensure that the participant referring another potential participant talks to them about the study, shares their experience participating, and secures their consent for their number to be shared with the research team. Although this may place a burden on the participant, it is essential in order to make sure that the referred participant is inclined to participate. Additionally, although the studies do not bring an immediate and direct benefit to the participants, it is essential to communicate that their participation is critical for generating data relevant to policymaking and decision-making in public health as part of the long-term purpose of conducting research. This strategy bore no specific additional cost.

### 3.3. Remote Strategy: Referrals from Social Workers and Pro-Migrant and Migrant-Led Organisations

All three projects also partly relied on referrals from social workers and pro-migrant and migrant-led organizations to recruit participants in Santiago and other regions, increasing the geographical reach of recruitment efforts. Before their beginning, each project was presented to social workers in healthcare centers in areas relevant to the study and where there is a concentration of international migrants and to pro-migrant and migrant-led organizations to seek their support in recruiting participants. Referrals were only carried out when the patient or member/beneficiary of the organization explicitly agreed to have their number shared with the research team. Additionally, none of the participating social workers and organization leaders or workers received monetary compensation, or any other kind of compensation, for recruiting participants, in compliance with ethical standards. In that sense, it also meant that recruitment costs remained low. The research team contacted the potential participants to provide them with further information, secure their formal consent unless they decided not to participate, and schedule the interview call. This strategy was feasible during mobility restriction and lockdown periods as it was integrated into ongoing, existing contact with potential participants.

The main challenge faced during the implementation of this strategy was that social workers and organization leaders expressed interest in participating in the study but did not always have the time to recruit participants as part of their regular activities, requiring a longer time frame to reach sample size and saturation. In that sense, recruitment must require as little work as possible for the social workers and pro-migrant and migrant-led organization leaders who agree to participate. Furthermore, when there is a potentially asymmetrical relationship between recruiters and potential participants, and although great care was taken in making sure that consent to share contact details is secured, the latter may accept to participate to preserve their relationship with the former; although, it is explicitly stated in the informed consent form that refusing to participate would not lead to negative consequences. Potential participants referred by social workers and organizations usually took part in the interview, making it a successful strategy. However, it left out populations experiencing more severe marginalization.

On the other hand, organization leaders may act as gatekeepers to protect members they perceive as especially vulnerable, and securing their support may be challenging. Therefore, developing a trusting, long-term, reciprocal, and non-instrumental relationship with pro-migrant and migrant-led organizations is critical and is usually initiated long before the start of the project. Additional ethical considerations must also be taken, and potential participants must be informed that refusal to participate or withdrawal from the study at a later stage will not be disclosed to, nor affect the relationship with, the person or organization who referred them. Additionally, they must be informed that the data collected and study results will not be shared in a way that will allow them to be identified personally in subsequent outputs. Finally, sharing research results and collaborative solutions improved trust.

### 3.4. Onsite Strategy: Vaccination Centres and Healthcare Centres

Although the first two projects were entirely carried out online and remotely, the third project was able to use onsite strategies. Based on increasing knowledge surrounding COVID-19 and prevention measures, the research team developed safe strategies aligned with government restrictions and recommendations to fulfill recruitment needs.

As infection rates kept falling at the time data collection started, making it safer for the research team to begin implementing onsite strategies, an agreement was made with the local health authorities to recruit and interview participants in vaccination centers and healthcare centers. Vaccination campaigns against both COVID-19 (initial doses and boosters) and the flu were ongoing, while routine medical attention had resumed in primary healthcare centers, making them ideal locations to recruit participants. Teams of two recruiters started going several times a week to healthcare centers and vaccination sites in the boroughs of interest in the project. In healthcare centers, recruiters were placed in waiting rooms and vaccination centers in the post-vaccination monitoring area, where the recently inoculated must remain for 15 to 30 min in case they experience an adverse reaction to the vaccine. Following an established protocol, they briefly presented the project and collected the contact details of those interested in participating. The recruiting team was trained to promote participation and participants’ commitment to the project by communicating the importance of hearing their voices as international migrants to generate cross-cultural dialogue with the health professionals and contribute to building together better access to healthcare in Chile. Furthermore, interviews were conducted onsite when the sanitary conditions allowed it, and participants agreed. In addition, administrative staff in healthcare centers agreed to mention the project to international migrants while they process their affiliation to the healthcare system or schedule an appointment. In addition, if the patient showed interest, they were directly referred to interviewers, further facilitating the onsite participation of international migrants and avoiding “singling out” international migrants in waiting rooms. No additional ethical constraint was identified.

Concerning geographical reach and cost, as the core research team was based in Santiago and the project was carried out in other regions, this strategy implied working with additional researchers living in these regions and travel expenses for the project coordinator, raising the cost of recruitment and data collection.

When the sanitary conditions did not allow the research team to conduct interviews onsite and only recruited participants, although many shared their contact details, some failed to respond when contacted later or refused to participate due to lack of time or interest. Others did not show up to the interview and did not reschedule. Some also provided a number that did not exist or belonged to someone else. Contacting potential participants recruited onsite by phone is labor-intensive and may lead to challenges for effective participation. Regarding the speed of recruitment and proportion of successful interviews compared to potential participants recruited, being able to interview onsite was a facilitator.

Additionally, the most vulnerable and at-risk may not access healthcare due to several barriers, and irregular migrants may not be aware of their right to be vaccinated in Chile, potentially leaving them out of the sample. Moreover, international migrants in Chile have reported lower perceived health needs than their local counterparts [7], which could influence the success of recruitment processes in healthcare centers and the characteristics of the sample recruited in these places.

### 3.5. Onsite Strategy: Community-Based Recruitment

Considering the challenge raised by recruitment within the health systems, especially not reaching those who do not access it, and given that the sanitary situation allows it, onsite recruitment at the community level was essential. Furthermore, concerning geographical reach and cost, as the core research team was based in Santiago and the project was carried out in other regions, this strategy presented the same characteristics as the other onsite strategy.

For the third project, community-based strategies were two-pronged. The first strategy involved working with recruiters who live in the areas of interest of the project. The second strategy consisted of recruitment by the research team in strategic territorial spots such as markets, small businesses and squares or parks, or during events organized by municipalities and non-governmental organizations tailored explicitly for international migrants.

The first challenge to community-based recruitment was that community recruiters had limited networks, outside of which they could not always engage with potential participants. Once this pool was exhausted, community recruiters usually withdrew from the project, leading to a high turnover. On the other hand, another significant challenge brought by community-based recruitment involving a research team external to the community was that high levels of distrust make it difficult, as outsiders, to successfully engage with members of the community who express suspicion regarding the legitimacy of the project. Furthermore, engaging with gatekeepers and community leaders yielded limited results for similar reasons. Additionally, they were usually women who juggled several roles and responsibilities at home, at work, and in the community and had little time to participate in the project. In some cases, they also experienced research fatigue.

More specifically, the boroughs that were the project’s focus did not historically receive international migrant populations, meaning that most were newly arrived, with an irregular migratory status, had limited support networks, and were usually highly mobile. In that sense, identifying formally or informally organized communities or associations was challenging. In addition, events aimed at the migrant population were not organized regularly, especially, as there was, at the time, a lot of uncertainty as to the future of the pandemic. However, overall, this last strategy successfully engaged with international migrants in a space they considered trustworthy and legitimate and led to interviews.

Community recruitment was successful under specific circumstances; however, considering the challenges described, it required time. Nevertheless, it allowed sample diversification, as it helped reach participants outside the healthcare system.

## 4. Discussion

To the best of our knowledge, this is the first report on qualitative research with hard-to-reach international migrant populations in Chile during the COVID-19 pandemic and one of the few at the regional and global levels. Although challenging, conducting public health research focused on hard-to-reach populations such as international migrants is vital to produce data relevant to policymakers and civil society for the inclusion of these populations in response to the pandemic. Long-term social unrest, closed borders, and increasingly restrictive migration policies in Chile have also fueled anti-migrant sentiment, and illegal deportations of international migrants were carried out frequently during the pandemic [37]. These, along with anti-migrant riots and violent attacks, especially in the northern regions of the country [38], have increased fear, mistrust, and marginalization, promoting mobility within Chile or returns to countries of origin, making including international migrants in research ever more critical, yet ever more difficult. Additionally, the COVID-19 pandemic has made research on topics other than the pandemic itself less prevalent, limiting opportunities to research other public health topics of urgent interest. This article presents the results of a reflection process carried out by the authors regarding three remote and two onsite strategies to recruit hard-to-reach international migrants to participate in qualitative health research in Chile during the COVID-19 pandemic. Seven categories of interest were identified: (i) feasibility during lockdown periods; (ii) speed of recruitment; (iii) geographical coverage; (iv) sample diversity; (v) proportion of successful interviews; (vi) ethical considerations; and (vii) cost.

In the context of the pandemic, researchers in Low and Middle-Income Countries (LMICs) have relied on community-based organizations and leaders, neighborhood health committees, established networks, existing programs targeting specific hard-to-reach population groups, and snowball sampling to facilitate recruitment of participants for public health research [22,39]. Regarding High-Income countries, research was conducted in Canada with resettled refugee youth. Recruitment relied on the personal networks of the lead investigator, who belonged to the same religious community as the participants, after the first strategy of working with local organizations failed as they were closed due to the pandemic [40]. While the first two projects presented in this article relied exclusively on remote engagement with study participants, the third project relied mainly on onsite recruitment in addition to minimal snowball sampling and referrals.

The existing literature shows that remote recruitment and data collection are successful in engaging with hard-to-reach populations, regardless of restrictions during the pandemic, because of the perception of increased confidentiality [41]. During the pandemic, it facilitated research while face-to-face engagement was limited [22]. More specifically, among remote recruitment strategies for hard-to-reach populations for public health or medical research, social media is prevalent and presents both advantages and disadvantages. The existing literature is divided with regard to its effectiveness as it depends on the characteristics of the population studied and whether other recruitment methods are available [42]. Most research teams use paid advertising on social media platforms such as Facebook and Instagram. Although it allows them to specifically target segments of the population and reach more people in less time than traditional recruitment methods [43], it can be costly [44], raise ethical and confidentiality concerns [45], and may not reach specific subgroups within the population of interest [42]. This may be especially true in LMICs, where research with vulnerable, marginalized, and hard-to-reach populations is constrained by connectivity. In this context, online studies may exclude entire subgroups among the most vulnerable within the hard-to-reach population. While Chile is considered a High-Income country by the World Bank [46] and most of its population has access to the Internet, challenges linked to unstable connection and low digital literacy limit its practical use among the most socioeconomically vulnerable, and the digital divide remains important [47,48].

Snowball sampling is also a prevalent remote recruitment method used with hard-to-reach populations because it is cost- and time-efficient and allows the recruitment of participants when there are multiple or very specific eligibility requirements [49]. However, although snowball sampling was successful in recruiting participants in the first two projects presented in this article, the third project faced the limitation that the international migrants surveyed are usually isolated enough that they do not have anyone to refer who meets the inclusion criteria. This suggests that although snowball sampling can help engage hard-to-reach populations, it may not be helpful with the “hardest-to-reach”.

Finally, establishing contact with potential participants through existing organizations, programs, or healthcare and social work professionals is another strategy to access hard-to-reach participants [39,50,51]. Prerequisites are establishing trust with individuals acting as gatekeepers and taking great care in preserving participants’ privacy and autonomy [40]. Time limitations may, however, hinder recruitment [52].

In the context of the projects presented in this article, conducting research entirely remotely was successful insofar as the sample sizes were relatively small in the first two projects as participants were usually recruited one by one among those who contacted the team via social media and those referred by social workers and organizations, and snowball sampling contributed to completing samples and reaching saturation. Conducting online recruitment for qualitative research with small samples including international migrants during the pandemic has been recommended in the existing literature [53]. Additionally, remote research allowed us to reach participants in the northern regions of the country without traveling, saving both the limited time and funds available and allowing us to conduct research with participants in quarantine facilities, where external visitors were not allowed. However, remote recruitment strategies present some limitations, and proactive, face-to-face contact, can be a more successful strategy for engaging with hard-to-reach populations [26,54]. Furthermore, conducting interviews via online platforms raised other challenges linked to developing trust with the participant and ensuring that the ethical requirements only to record sound and not videos of the interviews are strictly met. As our third project was implemented when mobility restrictions were gradually eased, focusing mainly on these methods was feasible [15].

### 4.1. Recommendations

Engaging with hard-to-reach migrant populations in times of adversity represents a critical challenge; however, relying on a flexible, proactive, interdisciplinary, and multicultural research team has been vital to carry out research projects during the COVID-19 pandemic successfully. In that sense, the need to constantly adapt recruitment strategies according to the evolving sanitary conditions, prevention measures, and mobility restrictions, called for iterative adjustments driven by constant feedback between research coordinators and the team in charge of recruiting and interviewing participants. Extremely adverse and volatile conditions mean things “may not go as planned”, and recruitment and data collection strategies, deliverables, and deadlines initially approved by funding organizations and ethics committees may need to be revised and adapted to changing conditions, requiring greater flexibility from these entities. Conducting qualitative research during the COVID-19 pandemic allowed us to collect valuable evidence from a population group that experienced new and exacerbated challenges. Additionally, it allowed us to experience different research methods in exceptional times and to raise good practices for recruiting hard-to-reach populations in that context. In this context, our main experience-based recommendations are presented in Box 1 below.

Box 1Experience-based recommendations for recruiting hard-to-reach migrant participants during a socio-sanitary crisis.General recommendationsConstitute a flexible, proactive, interdisciplinary, and multicultural research team.Allow for a flexible approach to recruitment according to evolving sanitary conditions, prevention measures, and mobility restrictions.Make iterative adjustments to strategies according to internal feedback.Whenever possible, seek to combine multiple remote strategies or remote and onsite strategies.Recommendations for social media recruitmentIf the study requires direct contact between the potential participant and the project coordinator, the latter must be ready and available to respond quickly to inquiries so as not to lose the interest of the potential participant.Sharing the invitation through institutional Facebook pages increases potential participants’ trust and harnesses pro-migrant organizations’ existing platforms. These organizations should have as much reach as possible both in terms of the number of followers and geographical coverage.Social media recruitment guarantees a flow of participants when onsite face-to-face recruitment is not allowed due to sanitary or other restrictions.Recommendations for snowball recruitmentAdequately communicates to the participants the importance of (i) participating in research and the specific project and (ii) referring other participants through basic scientific communication skills.Ensure that the participant asks for the referee’s consent to be contacted by the research team.Recommendations for recruitment via referrals from social workers and pro-migrant and or migrant-led organizationsBuild long-lasting relationships with institutions, social workers, and organizations.Ensure that the recruitment procedure is as light as possible to avoid placing an additional burden on workers and organization leaders.Additional ethical precautions must be taken as the referee may fear their relationship with the social worker or organization may be affected if they refuse to participate. They must also be informed that the data will be anonymized.Recommendations for recruitment in vaccination and healthcare centersRecruitment activities in healthcare centers and vaccination must be carried out in a way that does not disrupt the primary activities of the site, respect patient privacy and autonomy, and comply with all social distancing and prevention measures in place.Ideally, the recruiter contacts participants by phone the day the person was recruited to conduct the interview, mitigating non-response and refusal rates, and simplifying logistical and coordination aspects. This also minimizes the number of team members contacting the potential participant, thus avoiding confusion and misunderstandings.Interviewers should place several calls before marking a non-response.Introducing peer migrant recruiters improves trust and willingness to participate.Being able to interview participants onsite minimizes refusal and allows us to reach people with limited connectivity or low digital literacy.Recommendations for community-based recruitmentWorking with community members as recruiters builds trust around the project among the target communities and promotes participation.Involving migrant support offices from borough municipalities and non-governmental organizations with renowned trajectories also promotes trust from community leaders.Working with a multicultural team with recruiters and interviewers from different countries is key to engaging with potential participants from the international migrant community in a culturally and linguistically relevant manner.

### 4.2. Limitations and Strengths

Regarding the limitations, our results may be useful to other researchers in the field; however, this article does not represent an exhaustive guide on conducting research with hard-to-reach populations during pandemics or in general. Strategies that worked in the specific context of the projects presented may not be relevant in different contexts, require adaptations, and need long-term evaluation. We also acknowledge that since the studies included in the manuscript were carried out, most countries have stopped using lockdowns and mobility restrictions to prevent the spread of COVID-19. However, our results are relevant in several different ways. First, remote recruitment (both online and through snowball and referrals) may still be carried out in certain cases, for instance, to reach participants in places where the research team may not be able to go physically (environmental disaster, armed conflict, health hazard, or closed institutions such as quarantine facilities). Second, remote recruitment for qualitative research has previously been carried out by research teams with other hard-to-reach populations before the pandemic as it further increased confidentiality, as described by King, O’Rourke, and DeLongis [41]. Third, we recommend that recruitment processes include a combination of remote and onsite strategies to reach sample sizes and achieve sample diversity and saturation, which is relevant outside the pandemic. Fourth, in a similar line, the summary table in the result section, in addition to the narrated results for each strategy, can help researchers make decisions on how to recruit participants depending on the characteristics, context, and needs of their own research project. This article provides insight into different approaches to recruitment and their respective characteristics with regards to (i) feasibility during lockdown periods; (ii) speed of recruitment; (iii) geographical coverage; (iv) sample diversity; (v) proportion of successful interviews; (vi) ethical considerations; and (vii) cost. This information is relevant to promote innovation in qualitative research during social and sanitary crises or other adverse contexts.

## 5. Conclusions

This article presents the results of a reflection process carried out by the authors on three remote and two onsite strategies for recruiting international migrants to participate in qualitative public health research during the pandemic in Chile. Engaging hard-to-reach international migrants in public health research during the pandemic requires constantly adapting recruitment strategies to reach data saturation and when possible, combing remote and onsite recruitment is best. However, when face-to-face interaction is limited for sanitary reasons, using multiple online methods is useful for conducting research with hard-to-reach populations such as international migrants during the pandemic. Relying on not only Internet-based strategies allows the participation of people with limited access to the Internet and low-digital literacy. Iterative adaptation to the challenges of the pandemic and the subsequent social and health crisis is necessary to build rigorous scientific knowledge and target the specific needs of a constantly moving population.

## Figures and Tables

**Table 1 ijerph-20-01956-t001:** General description of the research projects.

Study Title	Study Design	Target Population and Sample Size	Location	Data Collection
International migrant communities’ vulnerability and resources during the COVID-19 pandemic in Chile: Building cross-cultural targeted coping strategies	Qualitative Semi-structured interviews	International migrants *n* = 30 Stakeholders *n* = 10	Metropolitan Region of Santiago and the Northern regions of Arica, Tarapacá, and Antofagasta	November–December 2020
International migrants in quarantine facilities in Chile during the COVID-19 pandemic: towards an ethical response to deeply rooted challenges	Qualitative Semi-structured interviews	International migrants in Chile who used quarantine facilities *n* = 17 Health professionals *n* = 10 Health authorities *n* = 5 Facility managers *n* = 1	Metropolitan Region of Santiago and the Northern regions of Arica y Parinacota, Tarapacá, and Antofagasta	December 2020–February 2021
Developing capacities for cross-cultural encounters between health teams and international migrants in Chile	Qualitative Semi-structured interviews and focus groups	International migrants (interviews) *n* = 48 International migrants (focus groups) *n* = 12 Health professionals (interviews) *n* = 40 Health professionals (focus groups) *n* = 12	Metropolitan Region of Santiago, Antofagasta Region, Valparaíso Region, and Araucanía Region	August 2021–August 2022

**Table 2 ijerph-20-01956-t002:** Strategies implemented in each research project.

Study Title	Remote Strategy: Social Media	Remote Strategy: Snowball Sampling	Remote Strategy: Referrals from Social Workers and Pro-Migrant and Migrant-Led Organizations	Onsite Strategy: Vaccination Centers and Healthcare Centers	Onsite Strategy: Community-Based Recruitment
International migrant communities’ vulnerability and resources during the COVID-19 pandemic in Chile: Building cross-cultural targeted coping strategies	X	X	X		
International migrants in quarantine facilities in Chile during the COVID-19 pandemic: towards an ethical response to deeply rooted challenges	X	X	X		
Developing capacities for cross-cultural encounters between health teams and international migrants in Chile		X	X	X	X

**Table 3 ijerph-20-01956-t003:** Summary of the qualitative evaluation of each strategy.

Strategy	Feasibility during Lockdown Periods	Speed of Recruitment	Geographical Reach	Sample Diversity	Proportion of Successful Interviews	Ethical Considerations	Cost
Remote strategy: social media	High	Low	High	Medium	Low	Low	Low
Remote strategy: snowball sampling	High	High	High	Low	Medium	Medium	Low
Remote strategy: referrals from social workers and pro-migrant and or migrant-led organizations	High	Low	High	Medium	High	High	Low
Onsite strategy: vaccination centers and healthcare centers	Medium	Medium	Medium	Medium	Medium	Medium	High
Onsite strategy: community-based recruitment	Low	Low	Medium	High	Medium	Medium	High

## Data Availability

All the data available is presented in the article.

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
