# Peer review of "Promoting the Participation of “Hard-to-Reach” Migrant Populations in Qualitative Public Health Research during the COVID-19 Pandemic in Chile"

_ijerph, 2023, doi:10.3390/ijerph20031956_

Round 1

Reviewer 1 Report (New Reviewer)

• In terms of grammar, the English language of the paper should be improved.

• The "Introduction" part of the study should be expanded, considering the research objectives, problems, and hypotheses. • An explicit statement of the objective(s) or question(s) the review addresses should be provided. • The full search strategies for all databases, registers, and websites should be presented, including any filters and limits used. • All outcomes for which data were sought should be listed and defined. Whether all results that were compatible with each outcome domain in each study were sought (e.g. for all measures, time points, analyses) should be specified. * How was the snowball sampling preferred to other techniques?

Author Response

Reviewer 2 Report (Previous Reviewer 2)

The authors have taken into consideration feedback from the earlier review.

The methods and results section has been expanded upon using tables and descriptions.

Recommendations have been formulated based on reflections from the results section.

The limitations and strengths of the study are clearly stated.

The article has presented the results of a reflective process on strategies to recruit international migrants for qualitative public health research.  This adds to the literature and has demonstrated that using appropriate strategies, international migrants can be reached for qualitative research.

Round 2

Reviewer 1 Report (New Reviewer)

Dear Authors

Thank you for this interesting work.

Revisions are sufficient.

This manuscript is a resubmission of an earlier submission. The following is a list of the peer review reports and author responses from that submission.

Round 1

Reviewer 1 Report

Thanks for the opportunity to review this interesting and significant paper. It's well known that
covid-19 affects our life.
- However, in my opinion, the methods authors used are relatively simple.

• I particularly appreciate the fact that the authors carefully review the specialized literature in the field, using recent references from top journals. Further on, I suggest the addition of the following references:
-Benham, J. L., Lang, R., Kovacs Burns, K., MacKean, G., Léveillé, T., McCormack, B., ... & Marshall, D. A. (2021). Attitudes, current behaviours and barriers to public health measures that reduce COVID-19 transmission: A qualitative study to inform public health messaging. PloS one, 16(2), e0246941.

-O’Sullivan, K., Clark, S., McGrane, A., Rock, N., Burke, L., Boyle, N., ... & Marshall, K. (2021). A qualitative study of child and adolescent mental health during the COVID-19 pandemic in Ireland. International journal of environmental research and public health, 18(3), 1062.

-Mureșan, G. M., Văidean, V. L., Mare, C., & Achim, M. V. (2022). Were we happy and we didn’t know it? A subjective dynamic and financial assessment pre-, during and post-COVID-19. The European Journal of Health Economics, 1-20.
-Shorey, S., Ang, E., Yamina, A., & Tam, C. (2020). Perceptions of public on the COVID-19 outbreak in Singapore: a qualitative content analysis. Journal of Public Health, 42(4), 665-671.

What's more, the differences (or similarities) and the significance of this paper compared with previous studies are not well discussed. Still, I think the authors should develop the last paragraph with the limits of their paper and emphasize how they intend to combat them in the future. Where do you go from here on? These suggestions will improve the quality of this manuscript and it will make the paper more suitable for being published within MDPI.

Reviewer 2 Report

This is an article of interest as migration continues worldwide. This article is of interest as it presents five strategies, lessons learnt and good practices of engaging with international migrants in Chile  to participate in qualitative public health research during the challenges of the COVID-19 pandemic. 

The information presented has been collected during the  implementation of 3 research projects with hard to reach migrant populations. The description of the three projects are presented clearly. However not all the projects state the sample size. one project states a recruitment of 30 participants. It will be good to get the sample size of the other studies

All studies were reviewed and approved by ethics committees. The projects samples were purposive. However it is thought that many undocumented migrants may be under represented in the studies as they may not have smart phones/ internet,Know people to refer or access to the health services. The sample may not represent all the different international migrants

Lessons learnt and good practices for each strategy are well presented. It is concluded that a combination of strategies  may produce the best outcomes.

Reviewer 3 Report

The work presents important research content that can contribute both to a better understanding of the characteristics of migrant populations in Latin America, as well as their epidemiological impact by regions.

A relevant demonstration of the use of social media and technological resources for methodological application of research in situations of difficulties in face-to-face communication with “hard-to-reach” migrant populations. Finally, a demonstration of iterative adaptation to the challenges posed by the COVID-19 pandemic and the subsequent social and health crisis in regions of South America, allowing for the construction of rigorous scientific knowledge and meeting the specific needs of a population in constant movement.
